# Robustness Disparities in Commercial Face Detection

**Samuel Dooley**
University of Maryland
sdooley1@cs.umd.edu

**Tom Goldstein**
University of Maryland
tomg@cs.umd.edu

**John P. Dickerson**
University of Maryland
john@cs.umd.edu

## Abstract

Facial detection and analysis systems have been deployed by large companies and critiqued by scholars and activists for the past decade. Critiques that focus on system performance analyze disparity of the system's output, i.e., how frequently is a face detected for different Fitzpatrick skin types or perceived genders. However, we focus on the robustness of these system outputs under noisy natural perturbations. We present the first of its kind detailed benchmark of the robustness of two such systems: Amazon Rekognition and Microsoft Azure. We use both standard and recently released academic facial datasets to quantitatively analyze trends in robustness for each. Qualitatively across all the datasets and systems, we find that photos of individuals who are *older*, *masculine presenting*, of *darker skin type*, or have *dim lighting* are more susceptible to errors than their counterparts in other identities.

## 1 Introduction

Face detection systems identify the presence and location of faces in images and video. Automated face detection is a core component of myriad systems—including *face recognition technologies* (FRT), wherein a detected face is matched against a database of faces, typically for identification or verification purposes. FRT-based systems are widely deployed [Hartzog, 2020, Derringer, 2019, Weise and Singer, 2020]. Automated face recognition enables capabilities ranging from the relatively morally neutral (e.g., searching for photos on a personal phone [Google, 2021]) to morally laden (e.g., widespread citizen surveillance [Hartzog, 2020], or target identification in warzones [Marson and Forrest, 2021]). Legal and social norms regarding the usage of FRT are evolving [e.g., Grother et al., 2019]. For example, in June 2021, the first county-wide ban on its use for policing [see, e.g., Garvie, 2016] went into effect in the US [Gutman, 2021]. Some use cases for FRT will be deemed socially repugnant and thus be either legally or *de facto* banned from use; yet, it is likely that pervasive use of facial analysis will remain—albeit with more guardrails than are found today [Singer, 2018].

One such guardrail that has spurred positive, though insufficient, improvements and widespread attention is the use of benchmarks. For example, in late 2019, the US National Institute of Standards and Technology (NIST) adapted its venerable Face Recognition Vendor Test (FRVT) to explicitly include concerns for demographic effects [Grother et al., 2019], ensuring such concerns propagate into industry systems. Yet, differential treatment by FRT of groups has been known for at least a decade [e.g., Klare et al., 2012, El Khiyari and Wechsler, 2016], and more recent work spearheaded by Buolamwini and Gebru [2018] uncovers unequal performance at the phenotypic subgroup level. That latter work brought widespread public, and thus burgeoning regulatory, attention to bias in FRT [e.g., Lohr, 2018, Kantayya, 2020].

One yet unexplored benchmark examines the bias present in a system's robustness (e.g., to noise, or to different lighting conditions), both in aggregate and with respect to different dimensions of the population on which it will be used. Many detection and recognition systems are not built in house, instead making use of commercial cloud-based "ML as a Service" (MLaaS) platforms offered by tech giants such as Amazon amd Microsoft. The implementation details of those systems are not

Submitted to the 35th Conference on Neural Information Processing Systems (NeurIPS 2021) Track on Datasets and Benchmarks. Do not distribute.

exposed to the end user—and even if they were, quantifying their failure modes would be difficult. With this in mind, our **main contribution** is a wide *robustness benchmark* of two commercial-grade face detection systems (accessed via Amazon's Rekognition and Microsoft's Azure face detection APIs). For fifteen types of realistic noise, and five levels of severity per type of noise [Hendrycks and Dietterich, 2019], we test both APIs against images in each of four well-known datasets. Across these more than 5,000,000 noisy images, we analyze the impact of noise on face detection performance. Perhaps unsurprisingly, we find that noise decreases overall performance, and that different types of noise impact, in an "unfair" way, cross sections of the population of images (e.g., based on Fitzgerald skin type, age, self-identified gender, and intersections of those dimensions). Our method is extensible and can be used to quantify the robustness of other detection and FRT systems, and adds to the burgeoning literature supporting the necessity of explicitly considering fairness in ML systems with morally-laden downstream uses.

## 2   Related Work

We briefly overview additional related work in the two core areas addressed by our benchmark: robustness to noise and demographic disparity in facial detection and recognition. That latter point overlaps heavily with the fairness in machine learning literature; for additional coverage of that broader ecosystem and discussion around fairness in machine learning writ large, we direct the reader to survey works due to Chouldechova and Roth [2018] and Barocas et al. [2019].

**Demographic effects in facial detection and recognition.**   The existence of differential performance of facial detection and recognition on groups and subgroups of populations has been explored in a variety of settings. Earlier work [e.g., Klare et al., 2012, O'Toole et al., 2012] focuses on single-demographic effects (specifically, race and gender) in pre-deep-learning face detection and recognition. Buolamwini and Gebru [2018] uncovers unequal performance at the phenotypic subgroup level in, specifically, a gender classification task powered by commercial systems. That work, typically referred to as "Gender Shades," has been and continues to be hugely impactful both within academia and at the industry level. Indeed, Raji and Buolamwini [2019] provide a follow-on analysis, exploring the impact of the Buolamwini and Gebru [2018] paper publicly disclosing performance results, for specific systems, with respect to demographic effects; they find that their named companies (IBM, Microsoft, and Megvii) updated their APIs within a year to address some concerns that were surfaced. Subsequently, the late 2019 update to the NIST FRVT provides evidence that commercial platforms are continuing to focus on performance at the group and subgroup level [Grother et al., 2019]. Further recent work explores these demographic questions with a focus on Indian election candidates [Jain and Parsheera, 2021]. We see our benchmark as adding to this literature by, for the first time, addressing both noise and demographic effects on commercial platforms' face detection offerings.

In this work, we focus on *measuring* the impact of noise on a classification task, like that of Wilber et al. [2016]; indeed, a core focus of our benchmark is to *quantify* relative drops in performance conditioned on an input datapoint's membership in a particular group. We view our work as a *benchmark*, that is, it focuses on quantifying and measuring, decidedly not providing a new method to "fix" or otherwise mitigate issues of demographic inequity in a system. Toward that latter point, existing work on "fixing" unfair systems can be split into three (or, arguably, four [Savani et al., 2020]) focus areas: pre-, in-, and post-processing. Pre-processing work largely focuses on dataset curation and preprocessing [e.g., Feldman et al., 2015, Ryu et al., 2018, Quadrianto et al., 2019, Wang and Deng, 2020]. In-processing often constrains the ML training method or optimization algorithm itself [e.g., Zafar et al., 2017b,a, 2019, Donini et al., 2018, Goel et al., 2018, Padala and Gujar, 2020, Agarwal et al., 2018, Wang and Deng, 2020, Martinez et al., 2020, Diana et al., 2020, Lahoti et al., 2020], or focuses explicitly on so-called fair representation learning [e.g., Adeli et al., 2021, Dwork et al., 2012, Zemel et al., 2013, Edwards and Storkey, 2016, Madras et al., 2018, Beutel et al., 2017, Wang et al., 2019]. Post-processing techniques adjust decisioning at inference time to align with quantitative fairness definitions [e.g., Hardt et al., 2016, Wang et al., 2020].

**Robustness to noise.**   Quantifying, and improving, the robustness to noise of face detection and recognition systems is a decades-old research challenge. Indeed, mature challenges like NIST's Facial Recognition Vendor Test (FRVT) have tested for robustness since the early 2000s [Phillips et al., 2007]. We direct the reader to a comprehensive introduction to an earlier robustness challenge due to NIST [Phillips et al., 2011]; that work describes many of the specific challenges faced by

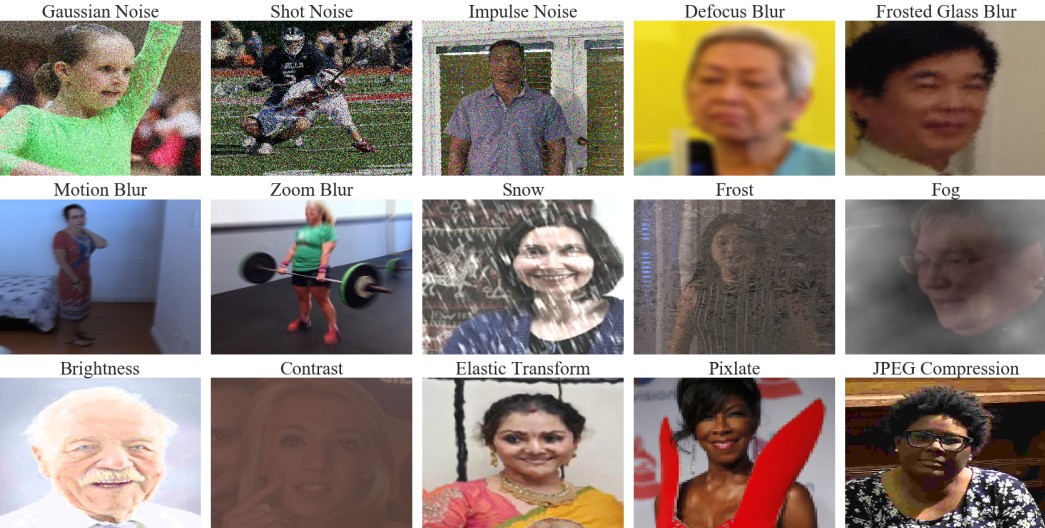

*Figure 1: Our benchmark consists of 5,066,312 images of the 15 types of algorithmically generated corruptions produced by ImageNet-C. We use data from four datasets (Adience, CCD, MIAP, and UTKFace) and present examples of corruptions from each dataset here.*

face detection and recognition systems, often grouped into Pose, Illumination, and Expression (PIE). It is known that commercial systems still suffer from degradation due to noise [e.g., Hosseini et al., 2017]; none of this work also addresses the intersection of noise with fairness, as we do. Recently, *adversarial* attacks have been proposed that successfully break commercial face recognition systems [Shan et al., 2020, Cherepanova et al., 2021]; we note that our focus is on *natural* noise, as motivated by Hendrycks and Dietterich [2019] by their ImageNet-C benchmark. Literature at the intersection of adversarial robustness and fairness is nascent and does not address commercial platforms [e.g., Singh et al., 2020, Nanda et al., 2021]. To our knowledge, our work is the first systematic benchmark for commercial face detection systems that addresses, comprehensively, noise and its differential impact on (sub)groups of the population.

## 3 Experimental Description

**Datasets and Protocol.** This benchmark uses four datasets to evaluate the robustness of Amazon AWS and Microsoft Azure's face detection systems. They are described below.

The Open Images Dataset V6 – Extended; More Inclusive Annotations for People (**MIAP**) dataset [Schumann et al., 2021] was released by Google in May 2021 as a extension of the popular, permissive-licensed Open Images Dataset specifically designed to improve annotations of humans. For each image, every human is exhaustively annotated with bounding boxes for the entirety of their person visible in the image. Each annotation also has perceived gender (Feminine/Masculine/Unknown) presentation and perceived age (Young, Middle, Old, Unknown) presentation.

The Casual Conversations Dataset (**CCD**) [Hazirbas et al., 2021] was released by Facebook in April 2021 under limited license and includes videos of actors. Each actor consented to participate in an ML dataset and provided their self-identification of age and gender (coded as Female, Male, and Other), each actor's skin type was rated on the Fitzpatrick scale [Fitzpatrick, 1988], and each video was rated for its ambient light quality. For our benchmark, we extracted one frame from each video.

The **Adience** dataset [Eidinger et al., 2014] under a CC license, includes cropped images of faces from images "in the wild". Each cropped image contains only one primary, centered face, and each face is annotated by an external evaluator for age and gender (Female/Male). The ages are reported as member of 8 age range buckets: 0-2; 3-7; 8-14; 15-24; 25-35; 36-45; 46-59; 60+.

Finally, the **UTKFace** dataset [Zhang et al., 2017] under a non-commercial license, contains images with one primary subject and were annotated for age (continuous), gender (Female/Male), and ethnicity (White/Black/Asian/Indian/Others) by an algorithm, then checked by human annotators.

For each of the datasets, we randomly selected a subset of images for our evaluation in order to cap the number of images from each intersectional identity at 1,500 as an attempt to reduce the effect of highly imbalanced datasets. We include a total of 66,662 images with 14,919 images from Adience; 21,444 images from CCD; 8,194 images from MIAP; and 22,105 images form UTKFace. The full breakdown of totals of images from each group can be found in Section A.1.

Each image was corrupted a total of 75 times, per the ImageNet-C protocol with the main 15 corruptions each with 5 severity levels. Examples of these corruptions can be seen in Figure 1. This resulted in a total of 5,066,312 images (including the original clean ones) which were each passed through the AWS and Azure face analysis systems. A detailed description of which API settings were selected can be found in Appendix C. The API calls were conducted between 19 May and 29 May 2021. Images were processed and stored within AWS's cloud using S3 and EC2. The total cost of the experiments was $9,887.17 and a breakdown of costs can be found in Appendix D.

**Evaluation Metrics.** Given that we aim is to study how corruptions to an image alter the commercial interpretation of that image, we valuate the error of the face systems. Additionally, none of the chosen datasets have ground truth face bounding boxes. Therefore, we can use the response from the clean image as a ground truth of sorts. Specifically, we take as ground truth the number of faces in an clean image and compare that to the number of faces detected in a corrupted image.

Our main metric is the relative error in the number of faces a system detects after corruption; this metric has been used in other facial processing benchmarks [Jain and Parsheera, 2021]. Measuring error in this way is in some sense incongruous with the object detection nature of the APIs. However, none of the data in our datasets have bounding boxes for each face. This means that we cannot calculate precision metrics as one would usually do with other detection tasks. To overcome this, we hand annotated bounding boxes for each face in 772 random images from the dataset. We then calculated per-image precision scores (with an IoU of 0.5) and per-image relative error in face counts and we find a Pearson's correlation of 0.91 (with $p < 0.001$). This high correlation indicates that the proxy is sufficient to be used in this benchmark in the absence of fully annotated bounding boxes.

This error is calculated for each image. The way in which this works is that we first pass every clean, uncorrupted image through the commercial system's API. Then, we measure the number of detected faces, i.e., length of the system's response, and treat this number as the ground truth. Subsequently, we compare the number of detected faces for a corrupted version of that image. If the two face counts are not the same, then we call that an error. We refer to this as the *relative corruption error*. For each clean image, $i$, from dataset $d$, and each corruption $c$ which produces a corrupted image $\hat{i}_{c,s}$ with severity $s$, we compute the relative corruption error for system $r$ as

$$rCE_{c,s}^{d,r}(\hat{i}_{c,s}) := \begin{cases} 1, & \text{if } l_r(i) \neq l_r(\hat{i}_{c,s}) \\ 0, & \text{if } l_r(i) = l_r(\hat{i}_{c,s}) \end{cases}$$

where $l_r$ computes the number of detected faces, i.e., length of the response, from face detection system $r$ when given an image. Often the super- and subscripts are omitted when they are obvious from context.

Our main metric, relative error, aligns with that of the ImageNet-C benchmark. We report mean relative corruption error ($mrCE$) defined as taking the average of $rCE$ across some relative set of categories. In our experiments, depending on the context, we might have any of the following categories: face systems, datasets, corruptions, severities, age presentation, gender presentation, Fitzpatrick rating, and ambient lighting. For example, we might report the relative mean corruption error when averaging across demographic groups; the mean corruption error for Azure on the UTK dataset for each age group $a$ is $mrCE_a = \frac{1}{15}\frac{1}{5}\sum_{c,s} rCE_{c,s,a}^{UTK,Azure}$. The subscripts on $mrCE$ will be omitted when it is obvious what their value is in whatever context they are presented.

Finally, we will also investigate the significance of whether the $mrCE$ for two groups are equal. For example, our first question is whether the two commercial systems (AWS and Azure) have comparable $mrCE$ overall. To do this, we will report the raw $mrCE$; these frequency or empiric probability statistics offer much insight into the likelihood of error. But we also indicate the statistical significance at $\alpha = 0.05$ determined by logistic regressions for the appropriate variables and interactions. For each claim of significance, regression tables can be found in the appendix. Accordingly, we discuss the odds or odds ratio of relevant features. See Appendix B for a detailed example. Finally, each

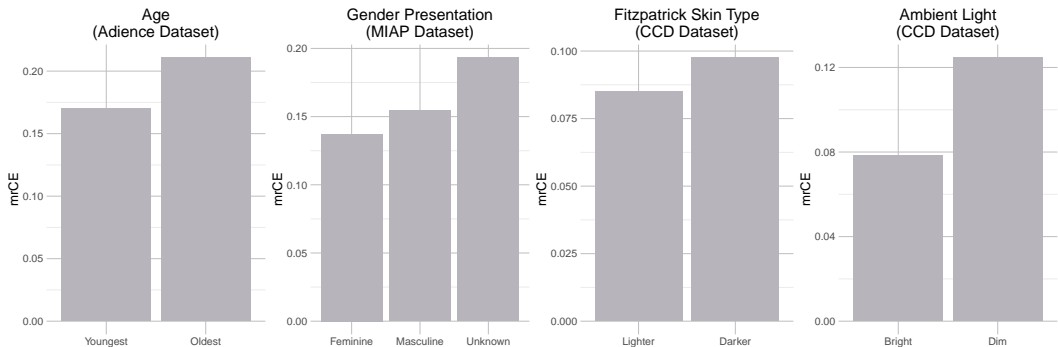

Figure 2: *There are disparities in all of the demographics included in this study; we show representative evidence for each demographic on different datasets. On the left, we see (using Adience as an exemplar) that the oldest two age groups are roughly 25% more error prone than the youngest two groups. Using MIAP as an exemplar, masculine presenting subjects are 20% more error prone than feminine. On the CCD dataset, we find that individuals with Fitzpatrick scales IV-VI have a roughly 25% higher chance of error than lighter skinned individuals. Finally, dimly lit individuals are 60% more likely to have errors.*

claim we make for an individual dataset or service is backed up with statistical rigor through the logistic regressions. Each claim we make across datasets is done by looking at the trends in each dataset and are inherently qualitative.

**What is not included in this study.** There are three main things that this benchmark does not address. First, we do not examine cause and effect. We report inferential statistics without discussion of what generates them. Second, we only examine the types of algorithmicaly generated natural noise present in the 15 corruptions. We speak narrowly about robustness to these corruptions or perturbations. We explicitly do not study or measure robustness to other types of changes to images, for instance adversarial noise, camera dimensions, etc. Finally, we do not investigate algorithmic training. We do not assume any knowledge of how the commercial system was developed or what training procedure or data were used.

**Social Context.** The central analysis of this benchmark relies on socially constructed concepts of gender presentation and the related concepts of race and age. While this benchmark analyzes phenotypal versions of these from metadata on ML datasets, it would be wrong to interpret our findings absent a social lens of what these demographic groups mean inside a society. We guide the reader to Benthall and Haynes [2019] and Hanna et al. [2020] for a look at these concepts for race in machine learning, and Hamidi et al. [2018] and Keyes [2018] for similar looks at gender.

## 4 Benchmark Results

We now report the main results of our benchmark, a synopsis of which is in Figure 2. Overall, we find that photos of individuals who are *older*, *masculine presenting*, *darker skinned*, or are *dimly lit* are more susceptible to errors than their counterparts. We come to these qualitative conclusions by quantitatively examining the trends of each dataset for each demographic. All four datsaets have age and gender labels. We see the bias against older individuals across all datasets. The bias against masculine presenting individuals is present in all datasets except UTKFace (which shows no bias). Skin type and lighting labels are only present in one dataset, CCD.

Below is a more detailed analysis with additional supporting tables and figures in the Appendix.

### 4.1 System Performance

Overall, AWS has fewer errors than Azure on corrupted data though the magnitude of the difference is small. The $mrCE$ for AWS is 12.298% whereas Azure is 12.338%, or 3% higher, but this is a Simpson's Paradox because when we look at each dataset, we see further nuance.

We plot the CTRs for each dataset and service in Figure 3; the difference between services is statistically significant for each dataset. For the Adience and MIAP datasets, Azure performs better

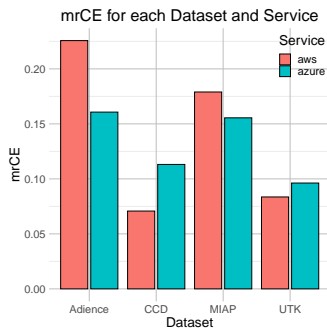

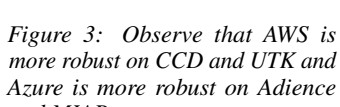

*Figure 3: Observe that AWS is more robust on CCD and UTK and Azure is more robust on Adience and MIAP.*

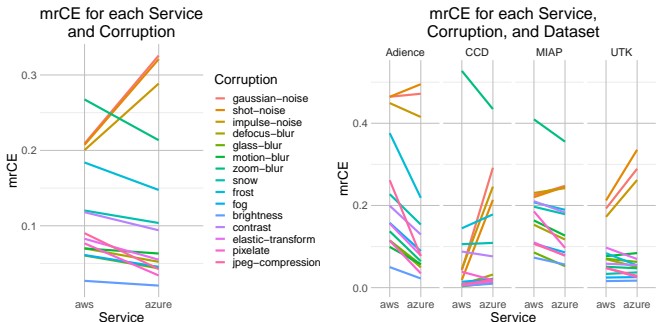

*Figure 4: A comparison of $mrCE$ for each commercial system and dataset where each line represents one of the 15 types of corruptions. (Left) depicts the robustness across all datasets whereas (right) depicts this for each dataset separately.*

than AWS. On Adience, Azure's $mrCE$ is 16.1% whereas AWS has $mrCE$ of 22.6%. The magnitude is less on MIAP; Azure has 15.6% and AWS has 17.9%.

Conversely, on the CCD and UTK dataset, Azure outperforms AWS. For the CCD dataset, Azure performs 60% worse than AWS (AWS $mrCE$ of 7.1% compared to Azure's 11.3%). The magnitude is less on UTKFace; AWS has 8.4% whereas Azure has 9.6%.

### 4.2 Noise corruptions are the most difficult

Recall that the ImageNet-C corruptions are broken into four different types: noise, blur, weather, and digital corruptions. We observe that the noise corruptions prove to be some of the most difficult corruptions for the commercial systems to handle. From Figure 4, we observe that in the AWS system, the three noise corruptions have the the second, third, and fourth most difficult corruptions (behind zoom blur). However, they are markedly the most difficult corruptions for Azure to handle. On the otherhand, Azure outperforms AWS on every other corruption. The difficulty of the noise corruptions echos that documented in the ImageNet-C experiments, though the comparative magnitude of the difficulty for these systems is significantly higher than what is previously documented.

When we examine the differences in the performance for each corruption across the different datasets, we see a continuation of the theme that the noise corruptions have relatively high $mrCE$. In every instance except one, Azure performs worse on the noise corruptions than AWS. For both commercial systems on Adience, the $mrCE$ values for the noise corruptions are above 40%. However, Azure preforms better than AWS on all other corruptions on the Adience Dataset.

The zoom blur corruption proves particularly difficult on the CCD and MIAP datasets, though Azure is significantly better than AWS (CCD: 52.7% for AWS and 43.5% for Azure; MIAP: 41.0% for AWS and 35.5% for Azure). We also note that all corruptions for all datasets and commercial systems are significantly differently from zero.

#### 4.2.1 Comparison to ImageNet-C results

Even though Hendrycks and Dietterich [2019] worked with the ImageNet dataset, we compare the findings from their paper to our experiments. We recreate Figure 3 from their paper with more current results for recent models since their paper was published, as well as the addition of our findings for AWS and Azure's face detection on our data; see Figure 8. This figure reproduces their metric, mean corruption error and relative mean corruption error. These differ from our metrics as they are defined as the raw error for each corruption, but normalized against the performance of AlexNet from the original paper. This is done so as to compare different models more fairly. The figure also shows the relative mean corruption error which is the difference between the raw error for each corruption and the raw error for the clean data. From this figure, we can conclude that our results are very highly in-line with the predictions from the previous data. This indicates that, even with highly accurate models, accuracy is a strong predictor of robustness.

We also examined the corruption-specific differences between our findings (with face data) and that of the original paper (with ImageNet data). We find that while facial datasets are most susceptible to

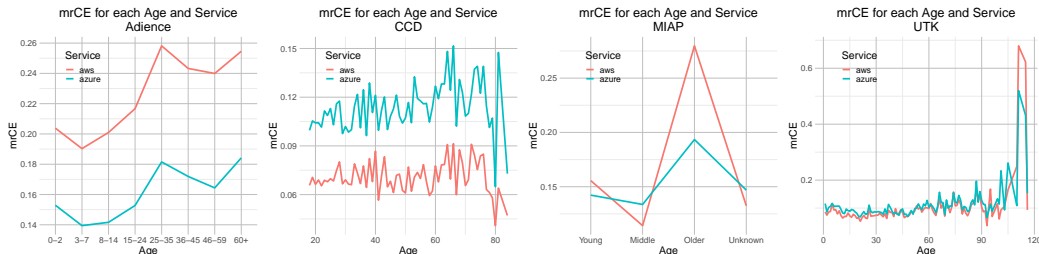

Figure 5: *Each figure depicts the $mrCE$ across ages. Each line depicts a commercial system (AWS is above Azure for Adience and MIAP). Age is a categorical variable for Adience and MIAP but a numeric for CCD and UTKFace. Observe the general trend of increased errors for older individuals.*

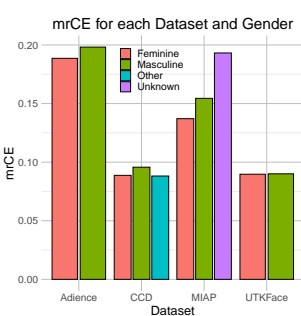

Figure 6: *Observe that on all datasets, except for UTKFace, feminine presenting individuals are more robust than masculine.*

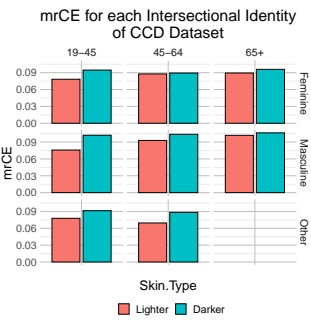

Figure 7: *In all intersectional identities, except for 45-64 females, darker skinned individuals are less robust than those who are lighter skinned.*

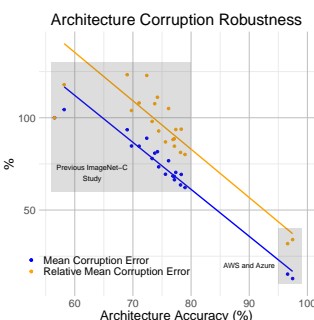

Figure 8: *Recreation of Figure 3 from Hendrycks and Dietterich [2019] with new results since their paper and the addition of our findings.*

noise corruptions, zoom blur, weather, etc, the ImageNet datasets are generally uniformly susceptible to corruptions with blurs and digital corruptions being the most difficult for them. This indicates that the face data have qualitative differences in their robustness susceptibility, indicating a need for further study.

### 4.3 Errors increase on older subjects

We observe a significant impact of age on $mrCE$. See Figure 5. In every dataset and every commercial system, we see that older subjects have significantly higher error rates. Recall that all four datasets have age metadata. Adience and MIAP have such data in groups. CCD and UTKFace have age data as a continuous variable.

On the Adience dataset, there is an interesting behavior where the second and third youngest age groups have the best performance with increases for younger and older age groups. There is then a spike in errors in the 25-35 age group which falls off slightly for the 36-59 groups and finally increases again for the oldest 60+ group. These two maximal groups have nearly 1:4 odds of error. This is compared to the youngest group which has 30% better odds (3:15).

For the MIAP dataset, the age disparity is very pronounced. Like the Adience dataset, we see a decrease in the likelihood of error moving from the youngest to the middle ages. However, we see a very large increase for the Oldest individuals. In AWS for instance, we see a 145% increase in error.

The CCD and UTKFace datasets have numeric age. Analyzing the regressions indicates that for every increase of 10 years, there is a 2.3% increase in the likelihood of error on the CCD data and 2.7% increase for UTKFace data. In Appendix E.4, we explore the interaction of Age and the corruptions.

### 4.4 Masculine presenting individuals have more errors than feminine presenting

Across all datasets except UTKFace, we find that feminine presenting individuals have lower errors than masculine presenting individuals. See Figure 6. On Adience, feminine individuals have 18.8% $mrCE$ whereas masculine have 19.8%. On CCD, the $mrCE$s are 8.9% and 9.6% respectively. On

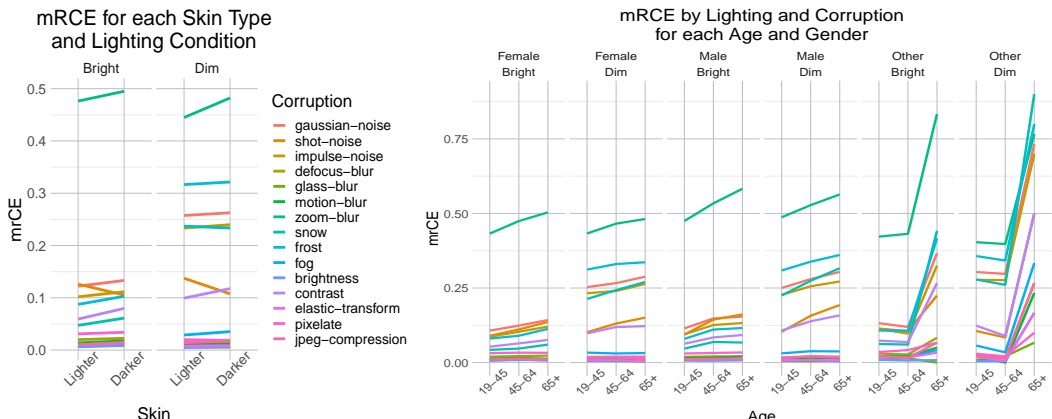

*Figure 9: (Left) $mrCE$ is plotted for each corruption by the intersection of lighting condition and skin type. (Right) the same is plotted by the intersection of age, gender, and lighting. Observe that for both skin types, all genders, and all ages, the dimly lit environment increases the error rates. Motion blur is the least robust corruption with frost, the three noises, and snow being the next worst across most intersectional identities.*

the MIAP dataset, the $mrCE$ values are 13.7% and 15.4% respectively. On the UTKFace, both gender presentations have around 9.0% $mrCE$ (non statistically significant difference).

Stepping outside the gender binary, we have two insights into this from these data. In the CCD dataset, the subjects were asked to self-identify their gender. Two individuals selected Other and 62 others did not provide a response. Those two who chose outside the gender binary have a $mrCE$ of 4.9%. When we include those individuals without gender labels, their $mrCE$ is 8.8% and not significantly different from the feminine presenting individuals.

The other insight comes from the MIAP dataset where subjects were rated on their perceived gender presentation by crowdworkers; options were "Predominantly Feminine", "Predominantly Masculine", and "Unknown". For those "Unknown", the overall $mrCE$ is 19.3%. The creators of the dataset automatically set the gender presenation of those with an age presentation of "Young" to be "Unknown". The $mrCE$ of those annotations which aren't "Young" and have an "Unknown" gender presentation raises to 19.9%. One factor that might contribute to this phenomenon is that individuals with an "Unknown" gender presentation might have faces that are occluded or are small in the image. Further work should be done to explore the causes of his discrepancy. In Appendix E.3, we explore the interaction of Gender and the corruptions.

### 4.5 Dark skinned subjects have more errors across age and gender identities

We analyze data from the CCD dataset which has ratings for each subject on the Fitzpatrick scale. As is customary in analyzing these ratings, we split the six Fitzpatrick values into two: Lighter (for ratings I-III) and Darker for ratings (IV-VI). The main intersectional results are reported in Figure 7.

The overall $mrCE$ for lighter and darker skin types are 8.5% and 9.7% respectively, a 15% increase for the darker skin type. We also see a similar trend in the intersectional identities available in the CCD metadata (age, gender, and skin type). We see that in every identity (except for 45-64 year old and Feminine) the darker skin type has statistically significant higher error rates. This difference is particularly stark in 19-45 year old, masculine subjects. We see a 35% increase in errors for the darker skin type subjects in this identity compared to those with lighter skin types. For every 20 errors on a light skinned, masculine presenting individual between 18 and 45, there are 27 errors for dark skinned individuals of the same category.

### 4.6 Dim lighting conditions has the most severe impact on errors

Using lighting condition information from the CCD dataset, we observe the $mrCE$ is substantially higher in dimly lit environments: 12.5% compared to 7.8% in bright environments. See Figure 9.

Across the board, we generally see that the disparity in demographic groups decreases between bright and dimly lit environments. For example, the odds ratio between dark and light skinned subjects is 1.09 for bright environments, but decreases to 1.03 for dim environments. This is true for age groups

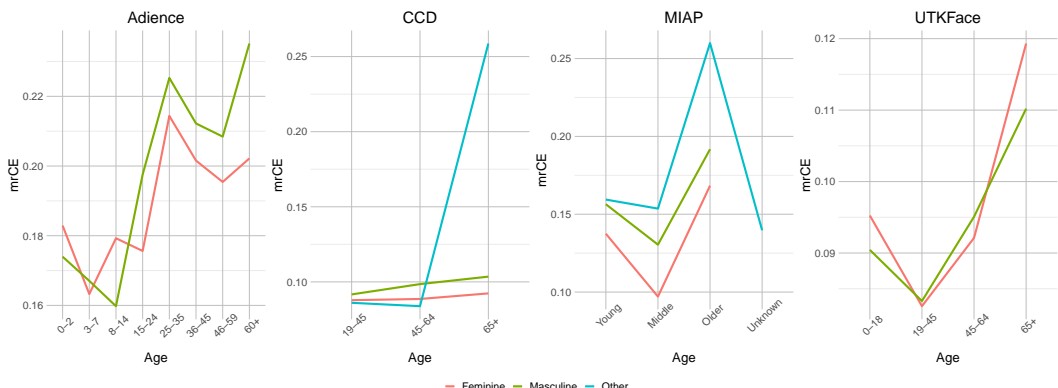

*Figure 10: For each dataset, the $mrCE$ is plotted across age groups. Each gender is represented and indicates how gender disparities change across the age groups.*

(e.g., odds ratios 1.183 (bright) vs 1.127 (dim) for 45-64 compared to 19-45; 1.138 (bright) vs 1.060 (dim) for Males compared to Females). This is not true for individuals with gender identities as Other or omitted – the disparity increases (1.104 (bright) vs 1.173 (dim) with Females as the reference).

In Figure 9 we observe the lighting differences for different intersectional identities across corruptions. We continue to see zoom blur as the most challenging corruption. Interestingly, the noise and some weather corruptions have a large increase in their errors in dimly lit environments across intersectional identities whereas many of the other corruptions do not.

### 4.7 Older subjects have higher gender error disparities

We plot in Figure 10 the $mrCE$ for each dataset across age with each gender group plotted separately. From this, we can note that on the CCD and MIAP dataset, the masculine presenting group is always less robust than the feminine. On the CCD dataset, the disparity between the two groups increases as the age increases (odds ratio of 1.048 for 19-45 raises to 1.135 for 65+). On the MIAP dataset, the odds ratio is greatest between masculine and feminine for the middle age group (1.395). The disparities between the ages also increases from feminine to masculine to unknown gender identities.

On the Adience and UTKFace datasets, we see that the feminine presenting individuals sometime have higher error rates than masculine presenting subjects. Notably, the most disparate errors in genders on these datasets occurs at the oldest categories, following the trend from the other datasets.

## 5 Gender and Age Estimation Analysis

We briefly overview results from evaluating AWS's age and gender estimation commercial systems. The detection model we evaluated for Azure does not provide age and gender estimates. Further analysis can be found in Appendices F and G.

### 5.1 Gender estimation is at least twice as susceptible to corruptions as face detection

The use of automated gender estimates in ML is a controversial topic. Trans and gender queer individuals are often ignored in ML research, though there is a growing body of research that aims to use these technologies in an assistive way as well [e.g., Ahmed, 2019, Chong et al., 2021]. To evaluate gender estimation, we only use CCD as the subjects of these photos voluntarily identified their gender. We omit from the analysis any individual who either did not choose to give their gender or fall outside the gender binary because AWS only estimates Male and Female.

AWS misgenders 9.1% of the clean images but 21.6% of the corrupted images. Every corruption performs worse on gender estimation than $mrCE$. Two corruptions (elastic transform and glass blur) do not have statistically different errors from the clean images. All the others do, with the most significant being zoom blur, Gaussian noise, impulse noise, snow, frost, shot noise, and contrast. Zoom blur's probability of error is 61% and Gaussian noise is 32%. This compares to $mrCE$ values of 43% and 29% respectively. See Appendix F for further analysis.

## 5.2 Corrupted images error in their age predictions by 40% more than clean images

To estimate Age, AWS returns an upper and lower age estimation. Following their own guidelines on face detection,[1] we use the mid-point of these numbers as a approximate estimate. On average, the estimation is 8.3 years away from the actual age of the subject for corrupted data, this compares to 5.9 years away for clean data. See Appendix G for further analysis.

## 6 Conclusion

This benchmark has evaluated two leading commercial facial detection and analysis systems for their robustness against common natural noise corruptions. Using the 15 ImageNet-C corruptions, we measured the relative mean corruption error as measured by comparing the number of faces detected in a clean and corrupted image. We used four academic datasets which included demographic detail. Adience, MIAP, and UTKFace have perceived age and gender metadata. CCD has subject provided age and gender responses as well as external ratings of skin type and ambient lighting conditions.

We observed through our analysis that there are significant demographic disparities in the likelihood of error on corrupted data. We found that older individuals, masculine presenting individuals, those with darker skin types, or in photos with dim ambient light all have higher errors ranging from 20-60%. We also investigated questions of intersectional identities finding that darker males have the highest corruption errors. As for age and gender estimation, corruptions have a significant and sizeable impact on the system's performance; gender estimation is more than twice as bad on corrupted images as it is on clean images; age estimation is 40% worse on corrupted images.

Future work could explore other metrics for evaluating face detection systems when ground truth bounding boxes are not present. While we considered the length of response on clean images to be ground truth, it could be viable to treat the clean image's bounding boxes as ground truth and measure deviations therefrom when considering questions of robustness. Of course, this would require a transition to detection-based metrics like precision, recall, and $F$-measure.

We do not explore questions of causation in this benchmark. We do not have enough different datasets or commercial systems to probe this question through regressions or mixed effects modeling. We do note that there is work that examines causation questions with such methods like that of [Best-Rowden and Jain, 2017] and [Cook et al., 2019]. With additional data and under similar benchmarking protocols, one could start to examine this question. However, the black-box nature of commercial systems presents unique challenges to this endeavor.

---

[1]`https://docs.aws.amazon.com/rekognition/latest/dg/guidance-face-attributes.html`

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
