# OpenReview forum: "Robustness Disparities in Commercial Face Detection"
_NeurIPS.cc/2021/Track/Datasets_and_Benchmarks/Round1 — Submitted to NeurIPS 2021 Datasets and Benchmarks Track (Round 1)_

### Official Review · Reviewer_3V1x · 2021-06-28
**Authors investigate the robustness disparities in commercial face detection. They observe some interesting findings. However, the core contribution of adding noise is not so convincing. Thus, the setup issue may occur in a lab environment rather than in a commercial face detection system.**

**Rating:** 6
**Confidence:** 4
**Clarity:** The paper is very well written.

**Strengths:**

1.	The paper is very well written and clearly conveys their ideas.
2.	It is the first systematic benchmark for commercial face detections that addresses and analyzes how the ``noise’’ affects the robustness of the facial detection system.
3.	It is an important research problem, and their experiments demonstrate intriguing results which are worthwhile for future investigation.

**Weaknesses:**


1.	The authors algorithmically generate the corruptions of images. Still, some of those noises do not look realistic, for example, the added snow and fog in Fig 1, and in fact, some of the noises in your corrupted images can be removed by many successful denoising algorithms. Such denoising algorithms are easily integrated into the current face detection system. So, could those denoising algorithms be applied to the facial detection systems and make them more robust to your noises? Please comment on this.

2.	Do you have the statistics about the number of images of different gender, ages, etc? When you contrast between two groups of pictures, do they keep balanced numbers of data? Do two contrasting groups of images have the same fractional amount of noises as different types of noises might influence the prediction? To draw a convincing conclusion, the above questions should be well discussed.

**Additional Feedback:**

Thanks for the authors' feedback. I have carefully checked the authors' feedback. They address my most of questions. I would like to increase my rating as the marginally above acceptance threshold.

**Correctness:**

The noise types were added by basic techniques, which is not very convincing. Because it’s difficult to assess how many of these things happen in real situations. Note that the added noises could be addressed by many successful denoising algorithms, which require further investigation.

**Documentation:**

Yes, they are available.

**Ethics:**

Yes.

**Relation To Prior Work:**

Yes.

**Summary And Contributions:**

The authors prob an important research problem in face recognition technology regarding how the noisy perturbations, including different ages, gender, skin color types, and lightening, could affect the robustness of current commercial face detection systems. They collect a subset of four public datasets from Open Images Dataset V6, The Causal Conversations Dataset, The Adience dataset, and the UTKFace dataset and corrupt the images by adding different types of noises. They use the noisy images to evaluate the robustness of Amazon AWS and Microsoft Azure’s face detection systems and demonstrate some interesting findings.

---

> ### Author Response · Authors · 2021-07-12
> **Precedence of using algorithmic corruptions**
>
> We’d like to thank you for your time and consideration in reviewing our paper. We are grateful for your perspectives and have updated our manuscript to reflect them.
>
> **Corruptions are algorithmically generated**
>
> It would certainly be ideal if there were a large corpus of data which had naturally occurring sensor noise which we could use for this study. However, no such dataset exists on the size and diversity with faces which we include here.
>
> Algorithmically generated noise models are widely studied and accepted in the robustness community as an important tool to measure, diagnose, and audit. There is debate about whether augmentation with synthetically generated noise can improve robustness as a mitigation technique; and it is generally thought that synthetic corruptions do not work well in general to improve robustness [1, 2, 3]. However, using algorithmically generated corruptions to measure robustness of these publicly available APIs is accepted, and at this time, the most effective and best way we have to audit such systems.
>
> *So, could those denoising algorithms be applied to the facial detection systems and make them more robust to your noises?*
>
> They certainly could! Without access to the model or procedure that AWS or Azure use in their face detection APIs, we can only speculate as to whether they already apply denoising prior to processing the image. With error rates so high for the noise corruptions, it seems to suggest that they do not; however, this is hard to believe that such well-resourced companies wouldn’t/couldn’t do this. It is reasonable though to believe that they don’t, rather making the assumption that the customer would input an image that isn’t noisy.
>
> Whether AWS or Azure apply denoising algorithms to user input images is ancillary to the main point of our paper. We find that these systems are more robust on some groups of people than others. Since these APIs are publicly available and widely used, understanding their failure modes as they exist today is extremely important for customers and citizens to know how they are impacted. Certainly some of these robustness problems could be addressed by denoising algorithms, but it is important to inform other academics and the public that they are likely being impacted right now by how the system works currently.
>
> **Convincing Conclusions**:
> *Do you have the statistics about the number of images of different gender, ages, etc?*
>
> We report the aggregate counts of individuals for each age, gender, skin type, and lighting condition for each dataset reported in Tables 1, 2, 3, and 4.
>
> *When you contrast between two groups of pictures, do they keep balanced numbers of data?*
>
> When making comparisons, it is not necessary to require the same number of images per group as our methods account for these differences and we do not violate any assumptions. When making quantitative conclusions, we do this by examining odds ratios in logistic regressions. Our standard logistic regressions adjust for the differences in the number of images in each group. Additionally, even our smallest group size is higher than conventional lower bound of 10 for logistic regressions.
>
> *Do two contrasting groups of images have the same fractional amount of noises as different types of noises might influence the prediction?*
>
> Yes, when making comparisons, the groups always have the same fraction of corruptions as well as the same type and fraction of corruptions. For example, if we are making a claim about group A and group B on the noise corruptions, then 93% of group B images are corrupted if and only if 93% of group A images are corrupted. Further, group A has only corruptions gaussian noise, shot noise, and impulse noise if and only if the same holds for group B (with the same breakdown).
> To draw a convincing conclusion, the above questions should be well discussed.
>
> We believe that the theoretical and statistical rigor of the logistic regression and the odds ratios  are convincing enough to believe any quantitative analysis which drives a quantitative conclusion. Every regression table used to derive a conclusion can be found in the Appendix for verification. As described starting at Line 161, we outline statistical significance is determined with $\alpha=0.05$ and many of our conclusions have much more power than that.
>
> [1] http://proceedings.mlr.press/v97/recht19a/recht19a.pdf
>
> [2] https://arxiv.org/abs/2006.16241
>
> [3] https://proceedings.neurips.cc/paper/2020/file/d8330f857a17c53d217014ee776bfd50-Paper.pdf

---

> > ### Comment · Reviewer_3V1x · 2021-07-15
> > **Comments on authors' response**
> >
> > Thanks. The authors have addressed my concerns. Thus, I decided to increase my final rating.

---

### Official Review · Reviewer_KtKY · 2021-06-28
**A good attempt at auditing face detection systems; however can be significantly improved**

**Rating:** 6
**Confidence:** 3
**Clarity:** Yes, the paper is well written with s…

**Strengths:**

Relevance to the broader research community: The authors clearly state what the paper does not address, in other words, they recognize some of the shortcomings. This shows that the authors have looked at the problem from a broad perspective, even if they could study only a part of the larger problem. This gives a good holistic picture to the readers of the paper.

Accessibility: The paper is generally presented well, with good structure and clarity

Ethical and social implications: The authors acknowledge that it would be wrong to interpret and generalize these results across different demographic groups.

Accountability: The authors have answered all the questions in the checklist, and provided other relevant information such as the cost incurred in conducting the experiments, detailed description of their experimental setting, etc.

Significance of the contribution: The authors consider an important problem and focus on an interesting setting within the problem context.

**Weaknesses:**

Significance and Contributions: There are quite a few concerns regarding the results presented, the datasets used in evaluation, the metrics proposed, and the analysis of the results themselves. Details are provided below.

1. The main claim on having "mixed results on system superiority" is questionable, especially given that on some subsets of the datasets, reverse trends were observed. These patterns remind one of the "Simpson's paradox", and therefore demands further analysis to understand which of the two opposing results is actually valid. In order to analyze this issue, it is necessary to understand the causal structure governing the various variables of interest in the dataset and the problem under consideration. As such, the validity of the claims therefore is questionable.
2. The conclusion made with respect to individuals who are older, masculine presenting, darker skinned, and those which are subjected to dim lights is perhaps not consistent across all the datasets studied in the paper. The datasets are not having similar characteristics, and so it is not clear how the results can be compared to arrive at a general trend. For e.g., the age distribution varies significantly across the datasets. So it is not clear as to what "older" means in this context when the authors make their claim.
3.  Some bias might have been introduced as a consequence of the experimental design-- both in terms of the number of images taken from the individual datasets, as well as the type of information available in each dataset.  Only about 8194 images are from MIAP as opposed to ~22000 from UTK and CCD datasets. While there has been some randomness in selection in order to counter for class imbalance, one cannot discount the fact that the distributions of these individual datasets are different. For e.g. Adience dataset has 8 age buckets, which is not necessarily the case in other datasets. This raises questions related to the intersectional results presented in the paper.
4. The evaluation metric proposed is not so very robust. The relative error in detecting the number of faces in a clean image and a corrupted image is not a reliable proxy---given that there can be many scenarios leading to false positives and false negatives. So, this cannot be a reliable indicator of a system's performance in detecting the face.
5. Some relevant prior work is not cited, the authors might want to explain how the present study takes the field forward in the context of these prior works.

Relevance to the broader research community :

In the context of the above concerns, the applicability of these results in a broader setting, and the benchmarks provided are not very robust.




**Additional Feedback:**

1. The authors may want to make note of a recent large scale face detection auditing experiment across diverse demographics. It might be interesting to analyze the results of their experimental setup in this problem setting to understand the feasibility of their method and generalizability of their results.
https://drive.google.com/file/d/1dvrqMzIsaeK2m2LxCVXWYu2K9WFZY32e/view
2. The authors may want to better motivate their problem by describing a practical scenario wherein it would be required to detect faces under such noise perturbations. While it is an interesting problem to study, it would be better appreciated/be more compelling with a practical use case.

**Correctness:**

There are issues with respect to the soundness of the claims made in the paper given that opposite trends were observed on data subsets and entire datasets across the two systems. Please see responses under "weakness" for details.
Also, there are issues related to the robustness of the evaluation metric as detailed earlier.
The experimental designs are somewhat biased in that the data distributions/information available in the datasets considered are different.  Please refer to the previous response for details.


**Documentation:**

Yes

**Ethics:**

The authors acknowledge that their results may not be generalizable across all demographics, and is limited to the experiments conducted.

Some points to ponder: There might be some concerns associated with "manipulating" the faces with such perturbations, for e.g. some individuals may object to their faces being distorted, some may become disturbed looking at the distorted faces (although this probability may be low).  Therefore, consent might be required from the concerned individuals ( whose images are in the dataset).

**Relation To Prior Work:**

The authors might want to consider the following relevant prior work and explain how their work is different, compare when possible, and describe how their work advances the field in the context of these prior works.

https://www.cs.cornell.edu/~shmat/shmat_wacv16.pdf





**Summary And Contributions:**

The authors evaluated the robustness of Microsoft Azure and Amazon Rekognition commercial face detection and analysis systems. The robustness analysis studied how adding different noise types with varying severity (fifteen types of realistic noise perturbations across five levels of severity) impacts the performance of these commercial systems in detecting the face.  For their analysis, the authors leveraged four existing face datasets, namely,  UTKFace, CCD dataset, Casual Conversation dataset, and  Open Images dataset V6.  Based on their experiments, the authors conclude that individuals who are older, masculine presenting, darker skinned, and subject to dim light conditions are more susceptible to errors than individuals belonging to other groups.

---

> ### Author Response · Authors · 2021-07-12
> **First response, more experiments to come (part 1)**
>
> We’d like to thank you for your time and thoughtful review. We have incorporated your suggestions and updated the manuscript to clarify any confusion. We particularly agree with your point about the validity of the metric. Although it has precedent in literature (in jian and parsheera [2021] for instance) there hasn’t been an analysis of its fidelity to detection metrics. We are currently conducting these experiments and will post the results when they are available later this week.
>
> **Responses to weaknesses**:
>
> **1** We fully agree with your point about Simpson’s paradox and believe we appropriately describe the Simpson’s paradox in Section 4.1. This is why we report both the overall performance of each system (12.298% for AWS and 12.338% for Azure) and we also report these same metrics for each dataset separately and show them in Figure 3. We have updated the subsection header to “System Performance”. Our very concern about the Simpson’s paradox is precisely what guides our careful review of each claim to ensure we do not have a Simpson’s paradox. As a clarifying point, the main claims of our paper come from the detailed analysis with regard to the demographic differences, described in Sections 4.3-4.7.
>
> **2**.  Your point is well taken and deserves further elaboration. Each claim we make *for an individual dataset or service* is backed up with statistical rigor through logistic regressions (see the Appendix). Each claim we make *across datasets* is done by looking at the trends in each dataset and are inherently qualitative. We have clarified this point in the body of the paper.
>
> For Gender, results are reported in Figure 6 with more details in Section 4.4. For every dataset, except for UTKFace, masculine presenting individuals have statistically significantly much higher error rates than feminine presenting individuals. We believe this is enough evidence to describe the general trend in a qualitative manner in the manuscript.
>
> Age presents a more difficult variable given its inconsistent reporting status across datasets, as you point out in your review. However, again we look at the qualitative picture which is painted by Figure 5 and see that across all datasets, the general trend is that the oldest individuals have significantly higher corruption errors than the youngest individuals. (Middle aged individuals appear to have relatively low corruption error.)
>
> Data on skin type and lighting condition are only present in one dataset and are thus reported.
>
> So we have evidence that the main claims are consistent across all datasets in the paper (with the caveat of UTKFace’s Gender finding).
>
> We apologize for the confusion stemming, presumably, from the wording of Figure 2. We have updated the figure and described it more in detail in the body of the paper in Section 4.
>
> **3**.  We believe that our sampling method combined with our rigorous statistical analysis mitigate the risk of selection bias when considering our experimental design and analysis protocol. When making comparisons, it is not necessary to require the same number of images per group as our methods account for these differences and we do not violate any assumptions. When making quantitative conclusions, we do this by examining odds ratios in logistic regressions. Our standard logistic regressions adjust for the differences in the number of images in each group. Additionally, even our smallest group size is higher than conventional lower bound of 10 for logistic regressions.
>
> We report the counts of images in each intersectional group in Tables 1-4 for each dataset.
>
> As for the differences in the ways the demographics are reported for each dataset, it is certainly unfortunate that these exist. However, we handled them in the most responsible way by performing all dataset-specific analysis with that dataset’s fidelity. Inter-dataset claims are made qualitatively and thus are more able to account for these differences.
>
> **4**. We agree that our metric is not perfect. None of these datasets have bounding boxes for faces, which would permit us to calculate precision, recall, and F-measures for these detection systems. We are currently annotating a portion of our dataset to perform an analysis of how highly correlated our metric is with the prue detection metrics. We will post another comment with an update of this analysis when it is completed.
>
> Given the state of the data, our metric is the best available to us and ther precedent for its use. In fact, the very interesting paper [1] you cited uses the same metric; they similarly did not have bboxes and opted to instead look at a measure of error defined by any difference in the number of detected faces.

---

> > ### Author Response · Authors · 2021-07-12
> > **First response, more experiments to come (part 2)**
> >
> > **5**. The work of Wilber et al [2] is very important and orthogonal to our work. Wilber et al use a much smaller dataset with many more types of occlusions and corruptions. When directly comparing the intersection of our two studies, we see that we both examine Gaussian noise and blurs. We both find that noise corruptions are harder than blurs. However, we find that in general, the systems of the present day are much better than the Facebook system was back in 2015. Our work is different in that it explores the important fairness question from an intersectional perspective. We advance the field with a large-scale study specifically designed to explore problems that are still understudied in the area of who is more disproportionately harmed and by how much by facial processing algorithms. Our study is the first to do that with facial detection and robustness questions. This will advance our understanding of where the field is currently so progress can be made and technologies adapted.
> >
> > **Points to Ponder**:
> > This is an important point and one that the community is coming to terms with at this moment. We see very good progress in this direction, for example the CCD dataset received full consent from the participants to explicitly be included in a machine learning dataset. We also will note that the concept of consent is very context dependent and informed consent is very hard to accomplish in most circumstances. We note that in ambiguous settings like much research with publicly scraped data, it is best practice to rely on the norms from which the data derived [3], which we believe we do.
> >
> > **1.4 Billion Missing Pieces?**
> > This is a very interesting work which aligns well with our experiments and we were very excited to see it presented at the workshop at CVPR. From our careful read of the paper, we have a nearly identical experimental set up (querying APIs with images of faces without boxes, asking intersectional questions) and identical metric. We do have data which we assume have more than one face in it, so our metric is specifically a generalization of theirs.
> > With respect to their results, they are not directly comparable as we are measuring errors on noise and they measure clean errors. However, both our papers show generally very low errors overall in the detection task. Since we do have the data on the uncorrupted images, in the next couple of days, we will analyze them to draw comparisons to their findings on age and gender prediction.
> >
> > [1] https://drive.google.com/file/d/1dvrqMzIsaeK2m2LxCVXWYu2K9WFZY32e/view
> >
> > [2] https://www.cs.cornell.edu/~shmat/shmat_wacv16.pdf
> >
> > [3] [Beyond the Belmont principles: Ethical challenges, practices, and beliefs in the online data research community](https://dl.acm.org/doi/abs/10.1145/2818048.2820078)

---

### Official Review · Reviewer_5NTr · 2021-07-01
**A face detection robustness benchmark with limited comparison with previous highly related work.**

**Rating:** 5
**Confidence:** 4
**Correctness:** Yes. Both of them are appropriate and…
**Clarity:** Yes. This paper is well written and e…

**Strengths:**

This work mainly focuses on the robustness of the commercial face detection system, through 15 corruptions. Besides, they also do a detailed analysis on the results, which might be useful and helpful for the owner of the APIs to do further improvement.

**Weaknesses:**

It would be much better if they could provide some cause and effect. Besides, the proposed metric and the whole process are very similar to the previous work [Hendrycks and Dietterich, 2019], since they used the 15 corruptions and similar metrics. They only mentioned twice in lines 42 and 96 in the manuscript, with a lack of comparison.

**Additional Feedback:**

Minors: 1. In lines 184 and 185, the authors claim that ‘Overall, AWS has fewer errors than Azure on corrupted data though the magnitude of the difference 185 is small. The mrCE for AWS is 12.3% whereas Azure is 3% higher than that. See Figure 3.’ However, the authors might miss reporting the whole results on two APIs.
2. Between lines 146 and 147, there is an equation to indicate the relative corruption error. But, I found three types of ‘r’. I suggest the authors could re-organize it.
3. There exist two kinds of ‘mrCE’ and ‘mRCE’ in the figure or manuscript. I suggest the author could only use just one.
4. In line 144, if possible, the authors might provide some examples of the system’s response to understand its length.
5. I suggest the authors could use just one image to show the 15 corruptions.

**Documentation:**

I think so if the authors could provide more details about 15 corruptions, the detailed parameters. Besides, the authors need to explain why 66,662 images could generate 5,066,312 images, not 4,999,650 images, through 75 times.

**Ethics:**

No. This work mainly focuses on the robustness of the face detection API and the experiments show some results and analysis of the existing systems.

**Relation To Prior Work:**

The authors should compare their work with [Hendrycks and Dietterich, 2019] since they used the 15 corruptions and similar metrics. They only mentioned twice in lines 42 and 96 in the manuscript. Maybe the authors need to do more comparisons.

**Summary And Contributions:**

In this paper, the authors present a benchmark to investigate the robustness of the commercial face detection system, such as Amazon Rekognition and Microsoft Azure. Using four public facial datasets, they firstly design a metric to measure the errors of the face systems, generate the corrupted images using 15 corruptions, and feed the clean and corrupted images to calculate the relative errors. They also report and analyze the benchmark results.

---

> ### Author Response · Authors · 2021-07-12
> **Update with Comparison to Hendrycks and Dietterich [2019]**
>
> We’d like to thank you for your time and helpful review. We especially appreciate the need to compare back to the original Hendrycks and Dietterich paper, which we have incorporated and found interesting results! We have updated the paper and highlighted new or modified text in blue.
>
> **Cause and effect**: We also believe that this would be a very interesting question to answer. We do not know of any form of causal reasoning which we could apply in this scenario given the complexity of the question and that we only have access to the API for the services. If you have suggestions, we certainly will explore them.
>
> **Comparison to results from Hendrycks and Dietterich [2019]**: We regret this important omission and have included a new comparison in Section 4.2.1. We note that the comparison is not entirely direct because we use different datasets but with the same corruptions. The original paper also uses a slightly different metric aimed to better compare between computer vision models. Therefore, our comparison is with their metrics. Nonetheless, we find that the mean corruption error of the AWS and Azure systems are entirely in line with previous results from Hendrycks and Dietterich [2019]. We have replicated Figure 3 from their paper with additional, recent results that they have posted on their github. Our addition of data confirms that accuracy has high predictive power over the model’s mean corruption error, even for high accuracy models like AWS and Azure face detection. We further find evidence that the types of corruptions that are the most difficult for AWS and Azure (noise corruptions, zoom blur, etc -- see Section 4.2) are generally not difficult for the models using the ImageNet dataset. This seems to suggest that the face datasets have higher susceptibility to these types of corruptions.
>
> **Specifics of corruptions**: We have included more details about the corruptions used in Appendix A.2 as well as a direct link to the python file which contains all the relevant parameters. Further, there are 5,066,312 images because we have 75 corruptions and 1 clean image so 66,662$\cdot$76 = 5,066,312 images.
>
> **Additional Feedback**:
> 1. We have updated lines 184 and 185 to include the raw number for each AWS and Azure. However, if we misinterpreted what you meant by “whole results”, we are happy to update.
> 2. Thank you for the suggestion. We have updated this to make it more clear.
> 3. Thank you for noting the inconsistency. We have updated this throughout.
> 4. We see the confusion and have adjusted the language in Lines 143 to address this. We additionally added Appendix B.1 to give explicit examples.
> 5. We debated whether to have Figure 1 include one image with the 15 corruptions or have different images. We ultimately decided to have different images because the former is included as Figure 1 of Hendrycks and Dietterich [2019]. We believe that the current format shows the diversity of images in our benchmark which is most important for our task.

---

> > ### Comment · Reviewer_5NTr · 2021-07-15
> > **Comments on authors' response**
> >
> > Thanks for the feedback. I have carefully read the reviews and the authors' feedback, as well as the updated manuscript. They address my most of questions. As for the cause and effect, it is quite challenging since we cannot access the training data and codes. For whole results, I mean the raw numbers of AWS and Azure (12.3% and 15.3%) don't occur in Fig. 3 but using 'see Figure 3' (minors).

---

### Author Response · Authors · 2021-07-14
**Summary of Revisions**

We thank the reviewers for their helpful comments which we believe have helped strengthen our submission. We have made two significant improvements. First, we report a comparison of our results with the original ImageNet-C paper (Figure 8). Our finding shows a significant trend that aligns remarkably well with the original paper and shows that system accuracy is a strong predictor of robustness, even with high accuracy models like in our study. Second, we showed that the object detection precision metric is highly correlated ($\rho$=0.91) with our metric (which is a proxy for precision since datasets with facial bounding boxes and demographics are small or do not exist). This alleviates any concern and shows that our metric reliably measures robustness bias as we believed.

We believe that our comparison to the ImageNet-C paper addresses the main concern of R1. The main criticism from R2 is that we do not examine causal reasoning for the trends we observe in our benchmark. While an interesting question, there is reliably no conceivable way to answer this question, and the reviewer provided no suggestions for what they would like to see. Additionally, causality analysis has no precedent in computer vision benchmarks, particularly facial recognition benchmarks pertaining to bias (see NIST’s FRVT benchmark [1] which explicitly rules out causality as a line of study). This causality criticism is orthogonal to a review of the merits of our contributions to a benchmark track. R3’s main criticism is that the algorithmic corruptions in our paper are unrealistic and therefore not worthy of study in a robustness bias benchmark. Based on the prolific citations for the ImageNet-C corruptions, the broad robustness community generally disagrees with this criticism, particularly given the imperative to understand robustness bias in commercial face detection system in spite of the dearth of naturally occurring noisy data with demographic labels.

[1] https://nvlpubs.nist.gov/nistpubs/ir/2019/NIST.IR.8280.pdf

---

### Decision · Program_Chairs · 2021-07-26

**Decision:**

Reject

**Comment:**

This paper presents an audit of two commercial facial analysis systems focusing on robustness to various types of image corruptions, with results further disaggregated along age, gender expression, and skin tone.

The method of analysis is similar to previous work of  Hendrycks and Dietterich. However, this paper focuses the audit on facial analysis systems rather than ImageNet and the results show that the same models exhibit different degrees of robustness to corruptions on ImageNet vs the facial datasets.

Overall this paper is well written and present results that are well within scope of this track and of interest to the NeurIPS community. However, the reviewers express concern with some of the empirical analysis and the conclusions that can be drawn from them. I believe there is an important contribution in this paper but that it would clearly be strengthened by another round of revisions that more fully address the reviewer's comments, specifically relating to the interpretation of overall trends across datasets and the inclusion of a more detailed error analysis. I encourage the authors to resubmit to the second round of the track.